# Peripheral line for vasopressor administration: Prospective multicenter observational cohort study for survival and safety

Adane Petros[1], Addisu Melkie[1], Kehabtimer Shiferaw Kotiso[2], Dawit Kebede[1], Chala Fekadu Oljira[1], Fitsum Assefa Gemechu[3]*, Hanan Yusuf[1], Sintayehu Abebe[1], Aschalew Ashagre[1], Amsalu Bekele[1], Andargew Yohannes[1], Eyob Kebede Etesa[4], Mohammed Bedru[5], Tewodros Haile Gebremariam[1]

1 Department of Internal Medicine, Addis Ababa University, College of Health Sciences, Addis Ababa, Ethiopia, 2 Department of Public Health, Werabe University, College of Health Sciences, Werabe, Ethiopia, 3 Addis Ababa University, College of Health Sciences, School of Medicine, Addis Ababa, Ethiopia, 4 East African Training Initiative, Addis Ababa, Ethiopia, 5 Cardiac Center Ethiopia, Addis Ababa, Ethiopia

* fitseaakfd@gmail.com

## Abstract

### Background

The placement of a central venous catheter (CVC) is associated with a high risk of central line-related bacteremia and mechanical complications. In a resource-constrained environment, the use of CVC is limited as a result of the complexity of the procedure and the need for a trained expert to place it. There is scant evidence on the feasibility, outcome, and safety of administering vasopressors using a peripheral venous catheter (PVC). Hence, our study, which is the first of its kind, evaluated the survival outcome, predictors, and safety of vasoactive drug administration via peripheral intravenous access in patients with circulatory shock.

### Methods

A prospective cohort study was conducted in the Emergency Department and ICU setting of selected public and private hospitals in Addis Ababa on selected 250 circulatory shock patients over 6 months for whom vasopressor was administered peripherally. Data was collected daily using ODK by study site coordinators and trained data collectors. Statistical analysis was done using STATA 14.1 using descriptive analysis, Kaplan Meir survival analysis, and cox regression analysis.

### Results

The median (IQR) age of the participants was 48.5 (35, 62). From circulatory shock causes, septic and cardiogenic shock accounted for 69.6% and 22.0% respectively. There were 3 extravasation injuries from the 250 patients that occurred exclusively

**Data availability statement:** All relevant data are available in a public repository at the following link: https://doi.org/10.5281/zenodo.16872086.

**Funding:** The funding for this research was acquired from Addis Ababa University, College of Health Sciences Research grant.

**Competing interests:** No authors have competing interests.

on patients that took vasopressor for more than 4 days yielding an event rate of 1.2% and 0.004/patient day. Mortality in circulatory shock patients is high with a 57.6% mortality rate in general and 67.8% in septic shock patients. A Cox regression analysis identified decreased baseline systolic blood pressure [AHR = 0.98; 95%CI: 0.95, 0.99], decreased urine output [AHR = 0.96; 95%CI: 0.93, 0.99], septic shock compared to cardiogenic shock [AHR = 0.48; 95%CI: 0.24, 0.94], and use vasopressor other than norepinephrine [AHR = 0.6; 95%CI: 0.39, 0.92] as independent predictors of death of circulatory shock patients with peripherally administered vasopressors.

## Conclusion

Our study found that peripheral venous catheter use for vasopressor administration in circulatory shock was associated with a low extravasation rate (1.2%), with all events occurring after more than five days, indicating it might be a safe alternative to CVC for short-term use in resource-limited settings. Findings support cautious prolonged use and consideration of central access when therapy exceeds a few days. This calls for randomized controlled trials to compare the safety and efficacy of vasopressor administration by PVC versus CVC.

## Introduction

Shock is a common condition in critical care, affecting about one-third of patients in the intensive care unit (ICU) that requires urgent intervention to decrease the high mortality associated with it [1]. It is defined as an acute or hyperacute physiological derangement, a systemic syndrome characterized by signs and symptoms of different organ hypo perfusion [2]. Circulatory shock is commonly caused by septic shock, a form of distributive shock which is the most common form of shock among patients in the ICU, followed by cardiogenic, hypovolemic shock, and obstructive shock [3]. Vasoactive medications (VMs) are often required to improve hemodynamic function in patients with shock [4]. They are usually given through central venous catheter (CVC). This is primarily out of concern that extravasation of peripheral intravenous (PIV) access may result in local tissue injury due to the vasoconstrictive effect of the VM. Evidence for this hypothesis was based on a systematic review for which case reports and case series contribute a significant share [5]. However, insertion of CVC is associated with a variety of mechanical complications and a risk of central line-associated bacteremia of around 15% [6]. Both central line-associated bacteremia and mechanical complications of CVC increase the length of hospital stay, mortality, and health care cost. The previous recommendation of early goal-directed therapy by Rivers et al described the use of a series of "goals" that included measurement of central venous pressure (CVP) and central venous oxygen saturation (Scvo2) which requires central venous catheter insertion for monitoring septic shock [7]. This approach has now been challenged as three large multicentric randomized control trials failed to show a mortality reduction [8–10]. Based on the three RCTs currently, major guidelines do not recommend central venous catheter insertion for septic shock intervention

monitoring over the usual bedside patient monitoring [4], which has limited the role of central venous catheter insertion in shock patients to the administration of vasoactive agents. The evidence on feasibility, outcome and safety of using vasoactive medications via Peripheral intravenous access for circulatory shock patients is mostly based on case reports and case series. There are limited prospective studies on the safety and outcomes of vasoactive medication administration via peripheral intravenous lines, and none have an adequate sample size or extend beyond a single institution. [11–13]. Despite the poor quality of existing evidence, vasoactive medication-related peripheral necrosis can lead to significant morbidity and negatively impact patient outcomes. A well-designed, prospective multicenter cohort study with a sufficient sample size is needed to evaluate the safety and efficacy of administering vasoactive medications through peripheral intravenous access in patients with circulatory shock, as current recommendations to use CVC for vasopressor administration are primarily based on case reports, case series, and small observational studies. Our study aims to assess the safety and outcome of administration of vasoactive medication via Peripheral venous line for circulatory shock patients.

## Methods and materials

### Study setting

The study was conducted in the Emergency Department and Intensive Care Units (ICUs) of public and private hospitals in Addis Ababa, Ethiopia, over a six-month period (May 1, 2021 – November 20, 2021). The participating hospitals with ICUs included Tikur Anbessa Specialized Hospital, Yekatit 12 Hospital, Eka Kotebe Hospital, Saint Peter Hospital, and Betezata Hospital.

### Study design and sampling

A Prospective cohort study design was used to assess the outcome and safety of administration of vasoactive medications via peripheral intravenous access for circulatory shock patients. The sample size was calculated using Epi info stat calc software for descriptive study with an event rate of 6% from Kamal et al study with an acceptable margin of error of 3% (reduced from the default value of 5% to increase the sample size) and 95% confidence interval(Z = 1.96). Design and clusters effect of 1 was used as this is a single center study with consecutive individual patients.

$$n = \frac{Z^2\ P(1-P)}{d2}$$

N = 241

 Adjusting for a non-response rate of 5%, the total sample size was 254. All subsequent patients in the study area who fulfilled the inclusion criteria were included.

 In this study, the observed extravasation rate was 1.2%, compared to the expected rate of 6% from prior literature. A post hoc power analysis, based on the observed absolute reduction and a two-sided alpha level of 0.05 was done. The result was 89% power to detect this effect. This highlighted that the study had sufficient power.

### Inclusion criteria

Any adult patients, age ≥ 18-year-old for whom vasoactive medications were administered through PIV access for more than 1 hour were included in the study.

### Exclusion criteria

- Non-traumatic amputation in any of the limbs
- Cardiac Arrest before initiation of vasoactive agents

## Operational definition

Safety of vasoactive medication administration via peripheral line was assessed in terms of extravasation injury and survival outcome. Minor safety considerations including changing peripheral access site because of malfunctioning were also included in the safety assessment.

**Circulatory shock** is defined in our study as a requirement for vasoactive medication to improve patients' blood pressure/ hemodynamics by the treating team.

Extravasation injury posing threat to the limb was considered a major safety concern.

**Extravasation injury** was defined as an injury in a peripheral access limb that includes irritation (pain), swelling, discoloration, and gangrene using the society of infusion nurses' infiltration scale. Vasoactive medications associated with peripheral line extravasation injury in this study include norepinephrine, epinephrine, phenylephrine, and dopamine.

**Peripheral intravenous access** is defined as venous access in the external jugular, upper arm, antecubital fossa, forearm hand, wrist, lower leg, and foot.

**Death** as a patient outcome was labeled whenever the patient dies before discontinuation of the vasoactive agent or in a 07- day follow-up period.

## Ethics approval and consent to participate

Informed written consent was collected from participants of the study or their next of kin if the patient was unable to give informed consent. Ethical clearance was obtained from the AAU CHS department of the Internal Medicine department review board. The safety and privacy of subjects were protected by using their de-identified information during the data collection and analysis process.

## Data collection techniques

A standardized case report format (CRF) was prepared to collect enrolled patient's data on demographics, indication for vasoactive medication, baseline risk factors for complications, PIV access site, the gauge of the catheter, Duration of Vasoactive medication, type of vasoactive medication, dilution of vasoactive medication, Infusion rate, Timing of complication and patient's outcome.

Follow-up for a patient receiving a vasoactive agent was discontinued under any of the following four conditions:

- If the patient dies

- If the Vasoactive agent is discontinued by the treating physician

- If the Patient develops extravasation injury

- When the study period is completed (07 days)

Data on all variables were collected using ODK by trained data collectors (residents, internists, and general practitioners) using the CRF prepared by the research team supervised by study site coordinators.

## Data quality and management

To ensure the quality of the data, pre-test data were collected on 5% of the sample size population from patients' medical records retrospectively in Tikur Anbessa Hospital, Addis Ababa University, and training was given to data collectors in Addis Ababa for one day before the survey to ensure consistency and reduce Intra and inter observation difference on the measurement of variables. The collected data were checked for completeness and consistency on each day of data collection. Daily supervision and monitoring were conducted by the assigned supervisors and principal investigators.

## Data processing and analysis

After data collection, survey responses obtained via ODK were converted to CSV and SPSS formats and then exported to Stata version 14.0 for analysis. Descriptive statistics, Kaplan-Meier survival plots, and Cox regression analysis were performed. Continuous variables were presented as mean (SD) or median (IQR), while categorical variables were reported as frequencies or percentages. Survival analysis was conducted to assess the mortality rate and complications associated with the administration of vasoactive medications via peripheral venous access in circulatory shock patients and to identify key determinants. A p-value of <0.05 was considered statistically significant in determining the impact on outcomes and complication rates.

## Result

### Clinical, vasopressor-related, and laboratory characteristics of the participants

In this study, 250 shock patients requiring vasopressor participated yielding a response rate of 98.4%. The median (IQR) age of the participants was found to be 48.5 [35−62] years. Nearly three-fourths (73%) of the patients were less than 60 years old. Among the participants, half of them were female 123 (49.2%). Approximately two-thirds (66.4%) of the study participants received their vasopressor in the ICU, while one-third (33.6%) received it in the emergency department. The commonest cause of circulatory shock was septic shock accounting for approximately 70% followed by cardiogenic shock which accounted for 22% of the patients. Among Comorbidities associated with a higher risk of extravasation, diabetes mellitus and hypertension were present in approximately 14.0% and 14.4% of patients, respectively. Nearly one fourth (23.6%) of the circulatory shock patients were COVID-19 positive (Table 1).

Table 2 shows vasopressor administration-related data of the study participants. Among the study participants, three-fourths (76.8%) took their vasopressor through the left extremity. Most (70%) of circulatory shock patients received their vasopressor from the peripheral venous catheter in the arm (42%) followed by the antecubital area (28%). Among the study participants, 58.8% and 23.6% took their vasopressor using 18-gauge and 20-gauge PVC. The most commonly used vasoactive agent was Norepinephrine (57.6%) followed by Epinephrine (41.6%). Around two-thirds (68.0%) of the circulatory shock patients achieved systolic blood pressure above 90-millimeter mercury (mmHg) within 24 hours of vasopressor initiation (Table 2).

The median Glasgow coma scale for study participants was 14 [IQR: 10–15] while the median systolic and diastolic blood pressure in mmHg before the initiation of vasopressor was 74.5 [IQR: 70–80] and 50 [IQR: 40–50] respectively. The median platelet count [$10^3$] of the study participants was 186 [IQR: 126–271.5] while the median absolute neutrophil count [$10^3$] was 9.6 [6.1, 15.3%]. The median urine output over 24 hours in ml was 400 [IQR: 300–1500]. Circulatory shock patients took norepinephrine during maximum infusion with a median dose of 0.25 [IQR: 0.2–0.5] mcg/kg/ ml. The patients also took the vasopressors for a median duration of 2.1 [IQR: 1.2–2.6] days (Table 3).

### Safety, survival outcome, and determinants

The mortality rate in circulatory shock patients was 57.6% with a 95% CI (51.3, 63.6). Septic shock patients have the highest mortality with 07 days mortality rate of 67.8% compared to 37.7% among cardiogenic shock patients.

Extravasation injury was rare among circulatory shock patients for whom vasopressor was administered with PVC with an incidence rate of 3 in 250 patients that took vasopressor for 750 days (1.2%) and 0.004 per/patient day. None of the extravasation injuries occurred in patients that took vasopressor for less than 05 days despite taking a maximum concentration of Norepinephrine of as high as 1 mcg/kg/min and a mean infusion rate of 0.3mcg/kg/min. The three patients with extravasation injury in our study are patients that took vasopressor via peripheral line in the hand for 05 days or more. All three patients subsequently died due to unresponsive circulatory shock with multi-organ failure. Two patients developed localized erythema and swelling of the hand, which did not require active management before their death. One patient

**Table 1. Clinical characteristics of vasopressor requiring shock patients admitted in hospitals with ICU capacity.**

| Variables (n = 250) | | n (%) |
|---|---|---|
| Age in year | 18–40 | 83 (33.2) |
| | 41–60 | 100 (40) |
| | >60 | 67 (26.8) |
| Sex | Male | 127 (50.8) |
| | Female | 123 (49.2) |
| COVID-19 Status | Yes | 59 (23.6) |
| | No | 163 (65.2) |
| | Unknown | 28 (11.2) |
| Comorbidities | Diabetes mellitus | 35 (14) |
| | Hypertension | 36 (14.4) |
| | Ischemic Heart Disease | 23 (9.2) |
| | Reynaud's Phenomenon | 1 (0.4) |
| | Smoking | 3 (1.2) |
| | Vasculitis | 2 (0.8) |
| | Stroke | 19 (7.6) |
| | Liver diseases | 11 (4.4) |
| | HIV/AIDS | 17 (6.8) |
| | SLE/APS | 2 (0.8) |
| Type of shock | Septic shock | 174 (69.2) |
| | Cardiogenic shock | 55 (22) |
| | Obstructive shock | 11 (4.4) |
| | Hypovolemic shock | 19 (7.6) |
| | Septic vs Cardiogenic not settled | 14 (5.6) |
| Was Systolic blood pressure > 90 achieved within 24 hours | Yes | 170 (68) |
| | No | 80 (32) |

experienced hand swelling and dark discoloration while on vasopressor therapy but died before the extent of the hand injury could be determined, despite receiving mechanical ventilator support and renal replacement therapy.

Before fitting the data to any model, the proportional hazards assumption was assessed for each covariate using the Kaplan-Meier survival plot. The results indicated that the survival curves were approximately parallel (Figs 1 and 2). However, visual inspection alone was not sufficient to confirm proportionality. Therefore, the proportional hazards assumption was further evaluated using the Schoenfeld residual test The test results were insignificant (p > 0.05) for all covariates, indicating that the assumption was met. Additionally, the global test was also insignificant, confirming that the proportional hazards assumption held (Table 4).

We have applied the cox model to see the effects of covariates on the outcome of interest (death) of patients. Variables with p-values of less than 0.25 from the univariable analysis were screened for multivariable analysis in the Cox-proportional hazard model. Variables include age, COVID-19 positivity, Sex, Baseline Systolic blood pressure, Urine output over 24 hours in ml, the Infusion rate of Norepinephrine during maximum infusion, serum albumin, Platelet in $10^3$, Absolute Neutrophil count in $10^3$, Type of shock, Creatinine, and Norepinephrine administration were a candidate for multivariable analysis. From these variables, some of the variables also were significant in multivariable cox regression at a p-value of 0.05.

For a 10 mmHg increase in systolic blood pressure, the hazard of death decreased by 20 percent keeping other covariates constant [AHR = 0.98; 95% CI (0.95–0.99)].

**Table 2. Vasopressor administration-related data of vasopressor requiring shock patients admitted in hospitals with ICU capacity.**

| Variables (n = 250) | | n (%) |
|---|---|---|
| Location of the venous access | External jugular | 12 (4.8) |
| | Upper arm | 6 (2.4) |
| | Antecubital fossa | 70 (28) |
| | Forearm | 106 (42.4) |
| | Hand | 47 (18.8) |
| | Wrist | 6 (2.4) |
| | lower leg | 1 (0.4) |
| | Foot | 2 (0.8) |
| Patients dominant hand | Right | 235 (94) |
| | Left | 15 (6) |
| Location of the venous access | left extremity | 192 (76.8) |
| | Right Extremity | 58 (23.2) |
| Intravenous catheter size in gauge | Green (18) | 147 (58.8) |
| | Pink (20) | 59 (23.6) |
| | Blue (22) | 43 (17.2) |
| | Yellow (24) | 1 (0.4) |
| Vasoactive agent used | Norepinephrine | 144 (57.6) |
| | Dopamine | 17 (6.8) |
| | Epinephrine | 104 (41.6) |
| Was SBP > 90 mmHg achieved within 24 hours | Yes | 170 (68) |
| | No | 80 (32) |

For those patients who were presented with cardiogenic shock, the hazard of death was decreased by 52% as compared to those patients who presented with septic shock keeping other covariates constant [AHR = 0.48; 95% CI (0.24, 0.94)]. For each 100 ml increase in urine output, the hazard of death decreased by 4% keeping the other covariates constant [AHR = 0.98; 95% CI (0.93–0.99)]. Norepinephrine use as compared to the use of other vasopressors decreased the hazard of death by 40% keeping the other covariates constant [AHR = 0.60; 95% CI (0.39–0.92)] (**Table 5**).

## Discussion

Our study is the first well-designed, prospective, multicenter study to evaluate the safety, survival outcomes, and determinants of poor prognosis associated with vasopressor administration via peripheral intravenous lines in patients with circulatory shock. Extravasation injury is rare among patients receiving vasopressor via peripheral line. Cox regression analysis identified baseline systolic blood pressure, urine output, type of shock, and norepinephrine use as independent predictors of survival in patients receiving peripherally administered vasopressors.

Current guidelines recommend CVC for vasopressor administration based on a systematic review of case reports and case series which reported limb-threatening extravasation injury with vasopressor administration via PVC [5]. However, our study aligns with findings from retrospective studies and protocolized observational studies with limited patient populations, demonstrating a low complication rate associated with peripheral vasopressor administration.[11,13]. Our patient cohort included a higher proportion of individuals at risk for extravasation injury based on established risk factors identified in previous studies. Several factors contributed to this increased risk, including a significant prevalence of Comorbidities that predispose patients to extravasation, variability in vasopressor concentrations and dosing, frequent placement of peripheral venous lines (PVLs) in high-risk locations, and the absence of a standardized protocol for managing extravasation events.

**Table 3. Measurement values of vasopressor requiring shock patients admitted in hospitals with ICU capacity.**

| Variables | Median (IQR) |
|---|---|
| GCS (Glasgow coma scale) | 14 (10, 15) |
| Systolic blood pressure before the initiation of vasopressor in mmHg | 74.5 (70, 80) |
| Diastolic blood pressure before the initiation of vasopressor in mmHg | 50 (40, 50) |
| Total WBC count * 10^3 | 11.05 (8, 16.7) |
| ANC (neutrophil count) *10^3 | 9.6 (6.1, 15.3) |
| ALC (lymphocyte count) *10^2 | 8.1 (3.9,12.3) |
| Hemoglobin in g/dl | 11.9 (9.6, 13.6) |
| Platelet count *10^3 | 186 (126, 271.5) |
| SGOT | 47 (26.9, 79.25) |
| SGPT | 33 (19, 59.95) |
| ALP | 120 (77.5, 189.5) |
| Total Bilirubin | 1.3 (0.6, 3.2738) |
| Direct Bilirubin | 0.47 (0.2, 1.58) |
| Albumin | 2.425 (2.1, 3.02) |
| Creatinine | 1.09 (0.67, 2.1) |
| Urea | 45 (24, 76.745) |
| PT | 16.8 (14.7, 21.4) |
| aPTT | 32 (25.3, 43.8) |
| INR | 1.45 (1.2, 1.8) |
| Urine output over 24 hr in ml | 400 (300, 1500) |
| Infusion rate of Norepinephrine during initiation in mcg/kg/min | 0.05 (0.05, 0.1) |
| The infusion rate of Dopamine during initiation in mcg/kg/min | 5 (5, 7.5) |
| Infusion rate of epinephrine during initiation in mcg/kg/min | 0.1 (0.1, 0.1) |
| The total duration of vasoactive agent administration in days | 2.1 (1.2, 2.6) |
| Infusion rate of Norepinephrine during maximum infusion in mcg/kg/min | 0.25 (0.2, 0.5) |
| The infusion rate of Dopamine during maximum infusion in mcg/kg/min | 10 (7.5, 27.5) |
| Infusion rate of epinephrine during maximum infusion in mcg/kg/min | 0.3 (0.2, 0.5) |

This contrasts sharply with previous studies on peripheral vasopressor administration, which were conducted on smaller patient populations and implemented strict safety protocols, including ultrasound-guided PVL insertion. Despite the substantial differences in risk factors between our study population and those in prior reports, we observed a comparable incidence of extravasation events.

Extravasation injury with peripheral line administration of vasopressors in circulatory shock patients in our study is rare with an incidence rate of 3 in 250 patients that took vasopressor for 750 days (1.2% patients or 0.004 per/patient days). This is in line with Cardenas-Garcia et al report and Delgado et al report of an extravasation event rate of 2% and 5% respectively, all of which were managed with local phentolamine injection, and led to no major complications in their retrospective and small sample size studies [12,14]. None of the extravasation injuries occurred in patients that took vasopressor for less than 5 days in our patient cohort. This suggests that vasoactive medications administration via peripheral line is safe for the duration of up to 4 days in terms of the risk of extravasation.

Unlike circulatory shock patients in other settings, our study population was relatively young, with a median age of 48 years and approximately 70% of patients under 60 years old. Despite being the youngest cohort compared to previous studies, the mortality rate remained high, particularly among patients with septic shock. The seven-day mortality rate was 52% for all circulatory shock patients and 67% for those with septic shock. This high mortality in septic shock

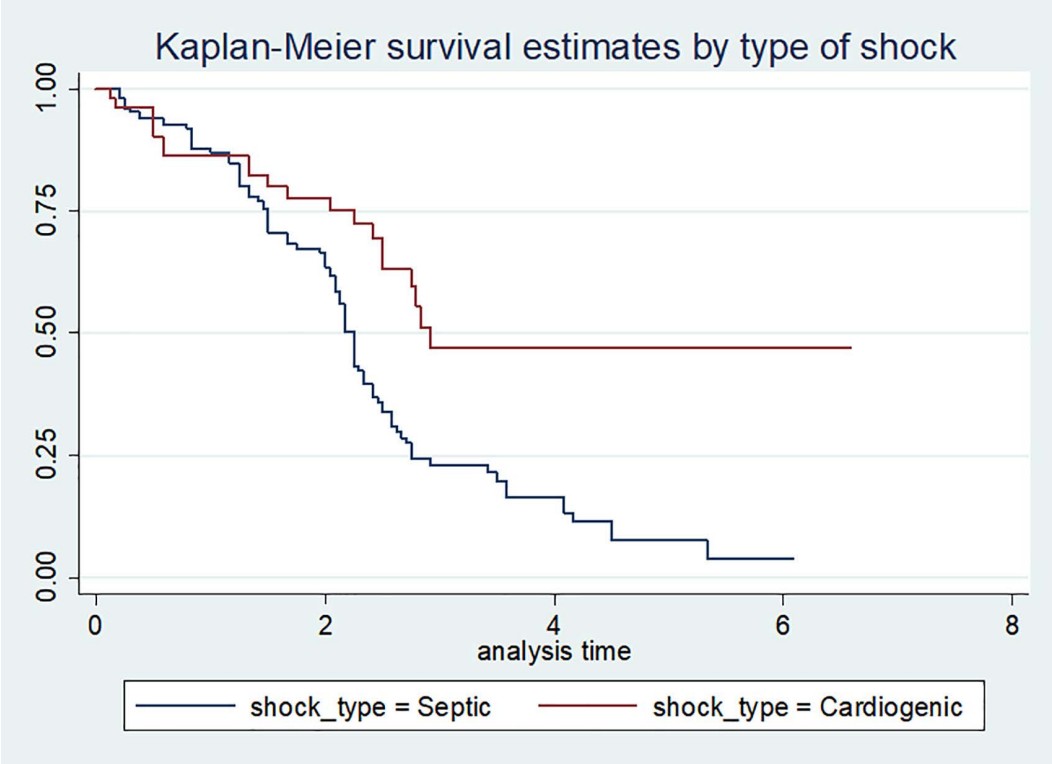

**Fig 1. Kaplan-Meier survival curve by type of shock among vasopressor requiring shock patients.**

patients in our cohort is in sharp contrast to the mortality rate of 20–50% in most septic shock cohorts from previous studies [7,15]. We concluded this difference is not related to peripheral line use in our study because cardiogenic shock patients in our study have a lower mortality rate than previous cohorts that use CVC. Besides, around 68% of our cohort patients have achieved SBP above 90 within 24 hours of vasopressor initiation which is comparable with previous studies. The mortality rate of cardiogenic shock patients in our cohort is 37.7% in sharp contrast to a reported mortality rate of around 70–75% from studies in high-income settings [16]. This is explained by the younger age of our patients and the predominant non-ischemic cause for the cardiogenic shock unlike the previous studies [17]. While the high mortality rate among septic shock patients in our study warrants further investigation to identify underlying causes, we attribute it primarily to inadequate general supportive care and delays in initiating septic shock management bundles. However, the persistently high mortality rate in septic shock patients, despite the younger age of our cohort, highlights the need for additional studies to explore contributing factors and for quality improvement initiatives to enhance patient outcomes.

Cox regression analysis identified baseline systolic blood pressure, urine output, type of shock, and norepinephrine use as independent predictors of outcomes in circulatory shock patients receiving peripherally administered vasopressors. These findings align with previous studies that have examined prognostic factors in septic and cardiogenic shock.

The hazard of death in circulatory patients in our cohort was lower with the use of norepinephrine. This is in line with previous studies that show lower mortality and arrhythmia complications with the use of norepinephrine compared with dopamine and epinephrine use [3,15]. The requirement of a higher dose of norepinephrine increased the hazard of poor outcomes in circulatory shock, finding congruent with previous cohort studies [18]. This could be explained by an underlying severe disease state which requires higher doses of norepinephrine.

                                                                                                                    9 / 13

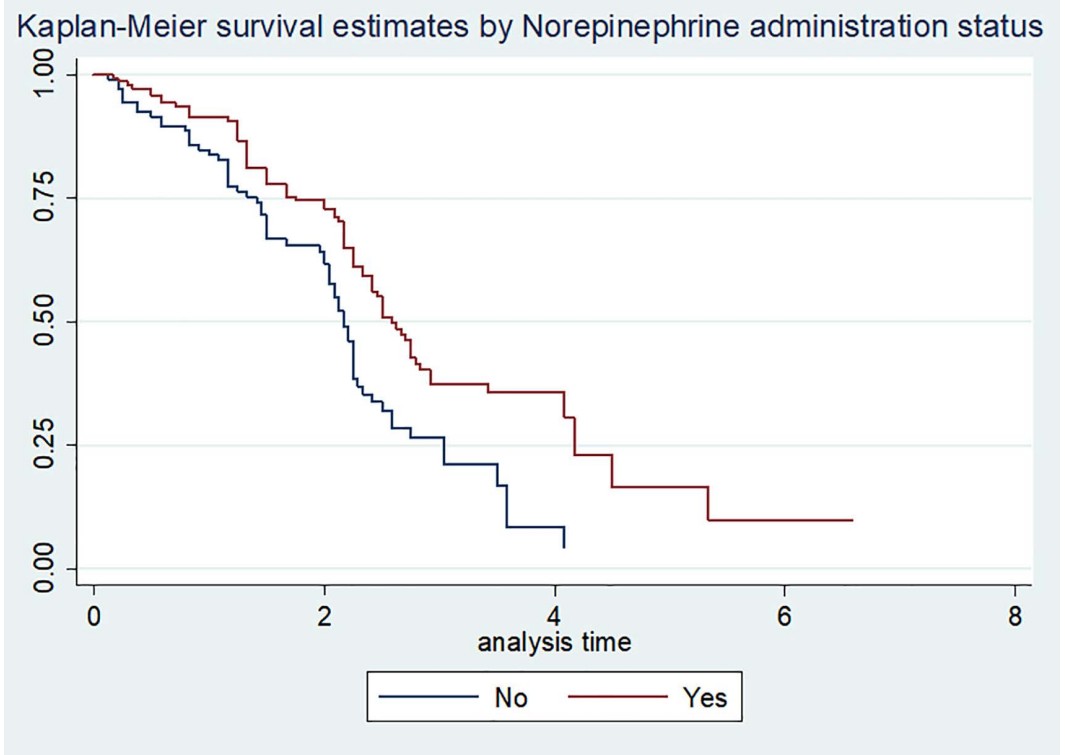

**Fig 2. Kaplan-Meier survival curve by norepinephrine administration status of vasopressor requiring shock patients admitted in hospitals with ICU capacity, Addis Ababa, 2021.**

**Table 4. Schoenfeld residual tests.**

| Covariates | ρ | χ² (1 df) | P value |
|---|---|---|---|
| Baseline systolic BP | 0.164 | 3.02 | 0.085 |
| Norepinephrine (Yes/No) | 0.139 | 2.15 | 0.146 |
| Urine output (100 mL) | 0.104 | 1.20 | 0.277 |
| Shock = Cardiogenic | −0.077 | 0.65 | 0.421 |
| Shock = Hypovolemic | 0.085 | 0.80 | 0.373 |
| Pulmonary embolism | −0.054 | 0.32 | 0.572 |
| Shock = Septic | 0.005 | 0.003 | 0.959 |

*Global test:* χ² = 8.14 (df = 7), p = 0.32.

Decreased urine output was associated with an increased hazard of poor outcomes. A similar finding was observed in previous studies on circulatory shock [19]. This could be explained by the fact that derangement in renal function is associated with severe circulatory shock with multiple organ failure.

A peculiar finding in our study is the lower hazard of mortality in cardiogenic shock patients in comparison to septic shock with a hazard ratio of AHR 0.48. This is in sharp contrast to the higher mortality rate of cardiogenic shock in comparison to septic shock in the western setup [3]. The lower mortality rate among cardiogenic shock patients in our cohort can be attributed to their younger age and differences in the underlying causes of cardiogenic shock. Notably, myocardial infarction-related cardiogenic shock, which is associated with poor outcomes, was less prevalent in our study population.

**Table 5. Bivariable and multivariable cox regression results of predictors of survival outcome among vasopressor requiring shock patients.**

| Variable | Death | | CHR (95% CI) | AHR (95% CI) | P-value |
|---|---|---|---|---|---|
| | Yes | No | | | |
| Baseline systolic blood pressure in mmHg | 70 (67, 80)¥ | 80 (70, 80)¥ | 0.80 (0.70, 0.92) | 0.98 (0.95, 0.99) | 0.038* |
| Type of shock | | | | | |
| Septic | 103 (67.8%) | 49 (32.2%) | 1 | 1 | |
| Cardiogenic | 20 (37.7%) | 33 (62.3%) | 0.35 (0.19, 0.64) | 0.48 (0.24, 0.94) | 0.033* |
| Hypovolemic | 10 (52.6%) | 9 (47.4%) | 0.69 (0.45, 1.34) | 0.78 (0.39, 1.54) | 0.472 |
| Card vs Septic | 7 (50%) | 7 (50%) | 0.54 (0.45, 0.91) | 0.42 (0.15, 1.18) | 0.099 |
| Pulmonary embolism | 4 (40%) | 6 (60%) | 0.43 (0.28, 1.20) | 0.54(0.18, 1.57) | 0.256 |
| Norepinephrine | | | | | |
| Yes | 79 (54.9%) | 65 (45.1%) | 0.46 (0.18, 0.84) | 0.60 (0.39, 0.92) | 0.019* |
| No | 65 (61.3%) | 41 (38.7%) | 1 | 1 | |
| 24-hour urine output in 100 ml | 4 (2, 10)¥ | 10 (4, 16)¥ | 0.87 (0.54, 0.97) | 0.96 (0.93, 0.99) | 0.012* |

**\*Statistically significant at a p-value of < 0.05 ¥ Median (Q1, Q3)**

## Limitations of the study

A key strength of this study is its multicenter, prospective cohort design, which included daily follow-up of circulatory shock patients over a seven-day period and incorporated survival analysis. This approach allowed for a detailed assessment of extravasation injury in terms of inpatient days and enabled precise documentation of the duration of vasopressor administration via peripheral venous catheters (PVCs) associated with an increased risk of extravasation.

However, several limitations should be noted. First, the absence of a control group, such as patients receiving vasopressors via central venous catheters (CVCs), limits direct comparisons and causal inference regarding the relative safety of PVC versus CVC administration. Due to severe resource limitations in the study setting, CVC insertion is rarely practiced, making a randomized controlled trial infeasible at present. Nevertheless, the current findings can serve as an important foundation for future RCTs on this topic.

Finally, the mortality rates observed in this study (57.6% overall and 67.8% among patients with septic shock) are notably higher than those reported in comparable studies. This discrepancy may be explained by several potential confounders, including late presentation of patients, limitations in ICU resources, higher baseline severity of illness, and delays between the decision to initiate vasopressor therapy and its actual administration. These factors should be considered when interpreting the study's findings.

## Conclusion

This study suggests that peripheral venous catheter (PVC) use for vasopressor administration in circulatory shock may be a practical alternative in resource-limited settings, particularly for short-term use. While the rate of extravasation was low (1.2%), all events occurred after more than five days of vasopressor administration, highlighting an increased risk with prolonged use. These findings indicate the need for clinicians to exercise caution and consider transitioning to central venous access when vasopressor therapy is anticipated to extend beyond a few days. Current guidelines, which mandate central venous catheter insertion in emergency settings, may warrant reassessment in light of these results, though the potential risks of long-term peripheral administration must be emphasized. Nevertheless, given the study's limitations, particularly the absence of a control group, further research is needed. Randomized controlled trials comparing the safety and efficacy of vasopressor administration via PVCs versus CVCs are essential to establish evidence-based recommendations.

## Author contributions

**Conceptualization:** Adane Petros, Addisu Melkie, Kehabtimer Shiferaw Kotiso, Dawit Kebede, Chala Fekadu Oljira, Tewodros Haile Gebremariam.

**Data curation:** Adane Petros, Addisu Melkie, Kehabtimer Shiferaw Kotiso, Dawit Kebede, Chala Fekadu Oljira, Tewodros Haile Gebremariam.

**Formal analysis:** Adane Petros, Addisu Melkie, Kehabtimer Shiferaw Kotiso, Dawit Kebede, Chala Fekadu Oljira, Fitsum Assefa Gemechu, Hanan Yusuf, Tewodros Haile Gebremariam.

**Funding acquisition:** Adane Petros, Addisu Melkie.

**Resources:** Adane Petros, Dawit Kebede.

**Supervision:** Adane Petros, Addisu Melkie, Kehabtimer Shiferaw Kotiso, Fitsum Assefa Gemechu, Hanan Yusuf, Tewodros Haile Gebremariam.

**Validation:** Addisu Melkie, Kehabtimer Shiferaw Kotiso, Dawit Kebede, Chala Fekadu Oljira.

**Visualization:** Adane Petros, Addisu Melkie, Kehabtimer Shiferaw Kotiso, Dawit Kebede, Chala Fekadu Oljira.

**Writing – original draft:** Adane Petros, Addisu Melkie, Kehabtimer Shiferaw Kotiso, Dawit Kebede, Chala Fekadu Oljira, Fitsum Assefa Gemechu, Hanan Yusuf, Sintayehu Abebe, Aschalew Ashagre, Amsalu Bekele, Andargew Yohannes, Eyob Kebede Etesa, Mohammed Bedru, Tewodros Haile Gebremariam.

**Writing – review & editing:** Addisu Melkie, Kehabtimer Shiferaw Kotiso, Dawit Kebede, Chala Fekadu Oljira, Fitsum Assefa Gemechu, Hanan Yusuf, Sintayehu Abebe, Aschalew Ashagre, Amsalu Bekele, Andargew Yohannes, Eyob Kebede Etesa, Mohammed Bedru, Tewodros Haile Gebremariam.

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
