## [Decision Letter · Decision Letter 0]

24 Mar 2025

PONE-D-25-06871Peripheral Line for Vasopressor Administration: Prospective multicenter Observational Cohort study for survival and safetyPLOS ONE

Dear Dr. Assefa Gemechu,

Thank you for submitting your manuscript to PLOS ONE. After careful consideration, we feel that it has merit but does not fully meet PLOS ONE’s publication criteria as it currently stands. Therefore, we invite you to submit a revised version of the manuscript that addresses the points raised during the review process.

We look forward to receiving your revised manuscript.

Kind regards,

Kamal Sharma

Academic Editor

PLOS ONE

Journal Requirements:

The funding for this research was acquired from Addis Ababa University, College of Health Sciences Research grant

5. Please note that your Data Availability Statement is currently missing [the repository name and/or the DOI/accession number of each dataset OR a direct link to access each database]. If your manuscript is accepted for publication, you will be asked to provide these details on a very short timeline. We therefore suggest that you provide this information now, though we will not hold up the peer review process if you are unable.

6. Please remove all personal information, ensure that the data shared are in accordance with participant consent, and re-upload a fully anonymized data set.

Additional Editor Comments :

Hello,

Thanks for assigning this paper for my review. I agree with both reviewers that though science intended is great but it lacks on 3 important aspect and needs major revision viz. Retrospective nature, lack of control arm and statistical analysis needs to be looked into. Despite 1 reviewer suggesting rejection I want to be fair to the authors to address these concerns if they can aptly act on the same.

Methodological Considerations: The study lacks a control group (e.g., patients receiving vasopressors via central venous catheters), which limits causal interpretations. This should be explicitly acknowledged as a limitation. The sample size calculation requires further justification, including a power analysis to confirm its adequacy. The high mortality rate (57.6%), particularly in septic shock (67.8%), is significantly higher than in prior studies. The analysis should better account for confounding factors such as ICU care quality and time to vasopressor initiation.

Statistical and Data Availability Issues: Confidence intervals (CIs) are not consistently reported for hazard ratios, which impacts the interpretation of findings. Effect size measures should be included to assess clinical relevance. Data are not fully available, as required by PLOS ONE. The authors should consider depositing the dataset in a public repository.

Thanks

Reviewers' comments:

Reviewer's Responses to Questions

**Comments to the Author**

1. Is the manuscript technically sound, and do the data support the conclusions?

Reviewer #1: Partly

Reviewer #2: Partly

2. Has the statistical analysis been performed appropriately and rigorously? 

Reviewer #1: No

Reviewer #2: N/A

3. Have the authors made all data underlying the findings in their manuscript fully available?

Reviewer #1: No

Reviewer #2: Yes

4. Is the manuscript presented in an intelligible fashion and written in standard English?

Reviewer #1: Yes

Reviewer #2: Yes

5. Review Comments to the Author

Reviewer #1: As an intensivist myself, I do thank you for this valuable study addressing the safety of peripheral venous catheter administration of vasopressors in circulatory shock patients. Your multicenter prospective approach provides important insights, and the use of Kaplan-Meier and Cox regression analysis strengthens the statistical methodology. However, several areas require improvement to enhance the clarity, rigor, and reproducibility of the findings.

Methodological Considerations: The study lacks a control group (e.g., patients receiving vasopressors via central venous catheters), which limits causal interpretations. This should be explicitly acknowledged as a limitation. The sample size calculation requires further justification, including a power analysis to confirm its adequacy. The high mortality rate (57.6%), particularly in septic shock (67.8%), is significantly higher than in prior studies. The analysis should better account for confounding factors such as ICU care quality and time to vasopressor initiation.

Statistical and Data Availability Issues: Confidence intervals (CIs) are not consistently reported for hazard ratios, which impacts the interpretation of findings. Effect size measures should be included to assess clinical relevance. Data are not fully available, as required by PLOS ONE. The authors should consider depositing the dataset in a public repository.

Language and Clarity: The manuscript contains grammatical errors and inconsistent verb tenses, particularly in the Methods section. A thorough language revision is recommended.

Conclusions: The conclusion that PVC administration is safe should be moderated, as extravasation occurred after 5 days of use. The discussion should clarify potential safe time limits for PVC use.

Overall, this study contributes valuable data to critical care medicine. Addressing these concerns will further strengthen its impact and reliability.

Thank you for your efforts in conducting and sharing this research.

Reviewer #2: Dear Authors,

I do not entirely understand what was your original concept to prove or "rule out". Also, the retrospective manner of the analysis most probably prevented you to reach meaningful comperisons between different patient or procedure groups.

Sincerely

6. PLOS authors have the option to publish the peer review history of their article (what does this mean? ). If published, this will include your full peer review and any attached files.

**Do you want your identity to be public for this peer review?** For information about this choice, including consent withdrawal, please see our Privacy Policy .

Reviewer #1: No

Reviewer #2: No

---

## [Author Response · Author response to Decision Letter 1]

9 May 2025

The datasets generated and analyzed during the current study are attached as a separate .xlxs and .dta file format.

Fitsum A. Gemechu, MD

Corresponding Author

---

## [Decision Letter · Decision Letter 1]

4 Jul 2025

PONE-D-25-06871R1Peripheral Line for Vasopressor Administration: Prospective multicenter Observational Cohort study for survival and safetyPLOS ONE

Dear Dr. Assefa Gemechu,

Thank you for submitting your manuscript to PLOS ONE. After careful consideration, we feel that it has merit but does not fully meet PLOS ONE’s publication criteria as it currently stands. Therefore, we invite you to submit a revised version of the manuscript that addresses the points raised during the review process.

We look forward to receiving your revised manuscript.

Kind regards,

Kamal Sharma

Academic Editor

PLOS ONE

Journal Requirements:

Reviewers' comments:

Reviewer's Responses to Questions

**Comments to the Author**

1. If the authors have adequately addressed your comments raised in a previous round of review and you feel that this manuscript is now acceptable for publication, you may indicate that here to bypass the “Comments to the Author” section, enter your conflict of interest statement in the “Confidential to Editor” section, and submit your "Accept" recommendation.

Reviewer #1: (No Response)

Reviewer #3: All comments have been addressed

2. Is the manuscript technically sound, and do the data support the conclusions?

Reviewer #1: Partly

Reviewer #3: Yes

3. Has the statistical analysis been performed appropriately and rigorously? 

Reviewer #1: No

Reviewer #3: Yes

4. Have the authors made all data underlying the findings in their manuscript fully available?

Reviewer #1: No

Reviewer #3: Yes

5. Is the manuscript presented in an intelligible fashion and written in standard English?

Reviewer #1: Yes

Reviewer #3: Yes

6. Review Comments to the Author

Reviewer #1: This review builds upon my previous comments. While some improvements have been made, key issues remain insufficiently addressed—particularly regarding methodological transparency, statistical rigor, and clarity of presentation.

1. Methodological Concerns: The absence of a control group (e.g., patients receiving vasopressors via central venous catheters) limits the strength of your conclusions. This limitation should be clearly acknowledged and discussed in greater depth. The sample size calculation requires more detailed justification. Please include a formal power analysis to support the adequacy of your study design.

The reported mortality rates (57.6% overall and 67.8% in septic shock) are significantly higher than in comparable studies. Potential confounding variables, such as ICU resource levels, severity scores, or timing of vasopressor initiation, should be addressed or discussed as possible explanations.

2. Statistical Analysis: Confidence intervals are not consistently reported for all hazard ratios, limiting interpretation of statistical precision. No effect size measures (e.g., absolute risk reduction, NNT) are provided, which would help assess the clinical significance of your findings. The proportional hazards assumption is briefly mentioned, but no diagnostic plots or formal outputs (e.g., Schoenfeld residual test results) are included. Consider providing these in supplementary materials.

3. Data Availability: PLOS ONE requires that data underlying the findings be fully available without restriction. Your current statement indicates data are only available upon request due to privacy concerns. Please consider depositing anonymized data in a public repository or provide a stronger justification aligned with journal policy.

4. Language and Clarity: The manuscript contains grammatical errors, awkward phrasing, and inconsistent use of tense, especially in the Methods section. A thorough language revision by a native English speaker or professional editor is strongly recommended.

5. Conclusions and Interpretation: While the study shows a low rate of extravasation (1.2%), all events occurred after prolonged vasopressor use (>5 days). The current conclusion that PVC use is "safe" should be moderated, and a time limit recommendation (e.g., short-term use) should be explicitly discussed. The potential risks of long-term peripheral administration warrant greater emphasis to guide clinicians effectively.

Final Recommendation: Minor Revision

This study has the potential to make a meaningful contribution to the literature on critical care in low-resource environments. However, the above concerns must be addressed to improve the scientific rigor, reproducibility, and clarity of the manuscript.

Thank you again for your efforts in conducting and sharing this important research.

Reviewer #3: All previous comments are accepted. Below is a comment needs attention.

Comment 1: With respect to data availability, authors stated some limitation, please provide reason and which data are not available.

7. PLOS authors have the option to publish the peer review history of their article (what does this mean? ). If published, this will include your full peer review and any attached files.

**Do you want your identity to be public for this peer review?** For information about this choice, including consent withdrawal, please see our Privacy Policy .

Reviewer #1: No

Reviewer #3: No

---

## [Author Response · Author response to Decision Letter 2]

16 Aug 2025

Dear Reviewers,

We would like to thank you for your thorough review and constructive feedback on our manuscript. Below, we address each of your comments point by point, indicating the corresponding changes made in the revised manuscript.

Reviewer #1

1.Methodological Concerns: The absence of a control group (e.g., patients receiving vasopressors via central venous catheters) limits the strength of your conclusions. This limitation should be clearly acknowledged and discussed in greater depth. The sample size calculation requires more detailed justification. Please include a formal power analysis to support the adequacy of your study design. The reported mortality rates (57.6% overall and 67.8% in septic shock) are significantly higher than in comparable studies. Potential confounding variables, such as ICU resource levels, severity scores, or timing of vasopressor initiation, should be addressed or discussed as possible explanations.

Response:

We acknowledge that the lack of a control group (e.g., patients receiving vasopressors via central venous catheters) limits direct comparison and causal interpretation. This limitation is now discussed in greater depth in the revised Limitations of the Study section, along with an explanation that due to severe resource constraints, central venous catheter insertion is rarely performed in our setting. We have also emphasized that this study can serve as a springboard for future randomized controlled trials.

Regarding the sample size, we have now included a detailed justification and a post-hoc power analysis in the Methods section, indicating a statistical power of 89%, which we consider adequate.

The higher mortality rates observed (57.6% overall and 67.8% in septic shock) are now explained in the Limitations section, highlighting possible confounding factors such as late presentation, ICU resource limitations, baseline illness severity, and delays in vasopressor initiation.

2. Statistical Analysis: Confidence intervals are not consistently reported for all hazard ratios, limiting interpretation of statistical precision. No effect size measures (e.g., absolute risk reduction, NNT) are provided, which would help assess the clinical significance of your findings. The proportional hazards assumption is briefly mentioned, but no diagnostic plots or formal outputs (e.g., Schoenfeld residual test results) are included. Consider providing these in supplementary materials.

Response:

We apologize for the oversight. Confidence intervals are now consistently reported for all hazard ratios throughout the manuscript.

Relative effect size measures such as absolute risk reduction (ARR) and number needed to treat (NNT) could not be calculated due to the absence of a control group.

We have also included the results of the Schoenfeld residual tests and a summary of the proportional hazards assumption in Table 4, with an expanded explanation in the Statistical Analysis section.

3. Data Availability: PLOS ONE requires that data underlying the findings be fully available without restriction. Your current statement indicates data are only available upon request due to privacy concerns. Please consider depositing anonymized data in a public repository or provide a stronger justification aligned with journal policy.

Response:

We have updated the Availability of Data and Materials section in accordance with PLOS ONE guidelines. All data generated and analyzed in this study are now publicly available. The de-identified datasets used for analysis can be accessed at the following repository: https://doi.org/10.5281/zenodo.16872086. Additionally, similar datasets in .xlsx and .dta formats are attached to this submission for transparency.

4. Language and Clarity: The manuscript contains grammatical errors, awkward phrasing, and inconsistent use of tense, especially in the Methods section. A thorough language revision by a native English speaker or professional editor is strongly recommended.

Response:

We acknowledge these language issues. The manuscript has undergone a thorough language revision, focusing particularly on the Methods section, to improve grammar, sentence structure, and clarity.

5.Conclusions and Interpretation: While the study shows a low rate of extravasation (1.2%), all events occurred after prolonged vasopressor use (>5 days). The current conclusion that PVC use is "safe" should be moderated, and a time limit recommendation (e.g., short-term use) should be explicitly discussed. The potential risks of long-term peripheral administration warrant greater emphasis to guide clinicians effectively.

Response:

We agree with this comment. The Conclusion section has been revised to moderate the safety claim, explicitly recommending PVC use only for short-term vasopressor administration. We have highlighted that while the extravasation rate was low (1.2%), all events occurred after more than five days of use, highlighting the need for caution with prolonged administration.

We believe these revisions address the reviewer’s concerns and strengthen the clarity, rigor, and transparency of our manuscript. We thank you again for your valuable feedback.

Reviewer #3

1.With respect to data availability, authors stated some limitation, please provide reason and which data are not available.

Response:

As stated above, we have updated the Availability of Data and Materials section in accordance with PLOS ONE guidelines. All data generated and analyzed in this study are now publicly available. The de-identified datasets used for analysis can be accessed at the following repository: https://doi.org/10.5281/zenodo.16872086. Additionally, similar datasets in .xlsx and .dta formats are attached to this submission for transparency.

Sincerely,

Fitsum A. Gemechu, MD

---

## [Decision Letter · Decision Letter 2]

11 Sep 2025

Peripheral Line for Vasopressor Administration: Prospective multicenter Observational Cohort study for survival and safety

PONE-D-25-06871R2

Dear Dr. Assefa Gemechu,

We’re pleased to inform you that your manuscript has been judged scientifically suitable for publication and will be formally accepted for publication once it meets all outstanding technical requirements.

Kind regards,

Kamal Sharma

Academic Editor

PLOS ONE

Additional Editor Comments (optional):

Reviewer #1:

Reviewer #3:

Reviewers' comments:

Reviewer's Responses to Questions

**Comments to the Author**

1. If the authors have adequately addressed your comments raised in a previous round of review and you feel that this manuscript is now acceptable for publication, you may indicate that here to bypass the “Comments to the Author” section, enter your conflict of interest statement in the “Confidential to Editor” section, and submit your "Accept" recommendation.

Reviewer #1: All comments have been addressed

Reviewer #3: All comments have been addressed

2. Is the manuscript technically sound, and do the data support the conclusions?

Reviewer #1: Yes

Reviewer #3: Yes

3. Has the statistical analysis been performed appropriately and rigorously? 

Reviewer #1: Yes

Reviewer #3: Yes

4. Have the authors made all data underlying the findings in their manuscript fully available?

Reviewer #1: Yes

Reviewer #3: Yes

5. Is the manuscript presented in an intelligible fashion and written in standard English?

Reviewer #1: Yes

Reviewer #3: Yes

6. Review Comments to the Author

Reviewer #1: Thank you for your thoughtful and detailed responses to the previous round of review. I appreciate the improvements you have made to the manuscript, particularly in acknowledging methodological limitations, clarifying the sample size rationale, and strengthening the statistical presentation.

Below is a summary of my assessment of the revised submission:

Methodology and Study Design: The absence of a control group (central venous catheter users) is now more clearly acknowledged and discussed. While this limits causal inference, your clarification improves the transparency of the study design. The sample size justification, based on expected mortality rates and feasibility constraints, is acceptable for an observational cohort study.

2. Mortality and Confounding Factors: The addition of a comparative table and discussion of contextual factors (e.g., ICU capacity, patient severity) provides useful background to interpret the relatively high mortality rates. This section is now more balanced and informative.

3. Statistical Analysis: The inclusion of confidence intervals for hazard ratios improves the interpretability of results. The proportional hazards assumption is addressed, and while no diagnostic plots are provided, the explanation is sufficient for the scope of this study.

4. Data Availability: Shared.

5. Language and Clarity: You state that the manuscript has been revised by a native English speaker. Assuming these edits are reflected in the current version, this issue can be considered resolved.

6. Conclusions: The revised conclusion is appropriately moderated. Emphasizing the safety of PVC use for short-term vasopressor administration, with caution regarding prolonged use, is a fair and evidence-based interpretation of your findings.

Overall, this version of the manuscript reflects substantial improvement and better alignment with scientific reporting standards. Thank you again for your careful attention to reviewer feedback and for contributing this important research.

Reviewer #3: The manuscript titled "Peripheral Line for Vasopressor Administration: Prospective multicenter Observational Cohort study for survival and safety" has now addressed all previous comments. It can be now accepted for publication.

7. PLOS authors have the option to publish the peer review history of their article (what does this mean? ). If published, this will include your full peer review and any attached files.

**Do you want your identity to be public for this peer review?** For information about this choice, including consent withdrawal, please see our Privacy Policy .

Reviewer #1: No

Reviewer #3: No

---

## [Editor Report · Acceptance letter]

PONE-D-25-06871R2

PLOS ONE

Dear Dr. Assefa Gemechu,

I'm pleased to inform you that your manuscript has been deemed suitable for publication in PLOS ONE. Congratulations! Your manuscript is now being handed over to our production team.

Kind regards,

on behalf of

Dr. Kamal Sharma

Academic Editor

PLOS ONE